

# A new voxel-based model for the determination of atmospheric-weighted-mean temperature in GPS atmospheric sounding

Changyong He[1], Suqin Wu[1], Xiaoming Wang[1], Andong Hu[1], Kefei Zhang[1,2]

[1]SPACE Research Centre, School of Sciences, RMIT University, Melbourne, VIC 3001, Australia
[2]School of Environment and Spatial Informatics, China University of Mining and Technology, Xuzhou, 221116, P.R.China

*Correspondence to*: Kefei Zhang (kefei.zhang@rmit.edu.au)

**Abstract.** The Global Positioning System (GPS) has been regarded as a powerful atmospheric observing system for determining precipitable water vapour (PWV) nowadays. One of the most critical variables in PWV remote sensing using GPS technique is the zenith tropospheric delay (ZTD). The conversion from ZTD to PWV requires a good knowledge of the
atmospheric-weighted-mean temperature ($T_m$) over the station. Thus the quality of PWV is affected by the accuracy of both ZTD and $T_m$. In this study, an improved voxel-based $T_m$ model, named GWMT−D, was developed and validated using global reanalysis data from 2010 to 2014 provided by NCEP-DOE Reanalysis 2 data (NCEP2). The performance of GWMT−D, along with other three selected empirical $T_m$ models, GTm−III, GWMT−IV and GTm_N, was assessed with reference $T_m$ derived from different sources – the NCEP2, Global Geodetic Observing System (GGOS) data and radiosonde
measurements. The results showed that the new GWMT−D model outperformed all the other three models with a root-mean-square error of less than 5.0 K at different altitudes over the globe. The new GWMT−D model can provide an alternative $T_m$ determination method in real-time/near real-time GPS-PWV remote sensing system.

## 1 Introduction

Water vapour (WV) is a major component of Earth's atmosphere and plays a vital role in global atmospheric radiation,
energy equilibrium and hydrological cycle (Wang et al., 2007). Since the Global Positioning System (GPS) became fully operational in 1994, it is possible to use GPS measurements to retrieve precipitable WV (PWV) in the atmosphere (Duan et al., 1996). The main advantage of using GPS to derive PWV is its high quality and availability of all time under all-weather condition with a global coverage. This feature is significantly advantageous for meteorological applications such as predicting short-term rainstorms and rainy seasons (Song et al., 2003;Zhang et al., 2007) and the monitoring of severe
weather events including thunderstorms, hail storms, strong winds and hurricanes (Choy et al., 2001;Zhang et al., 2015).

PWV is defined as an equivalent height of a column of liquid water. GPS-derived PWV, i.e. GPS-PWV, is converted from the zenith tropospheric delay (ZTD) estimated from GPS measurements. The GPS-PWV can be used for inter-comparisons among radiosonde, WVR (WV radiometer), MODIS (Moderate-Resolution Imaging Spectroradiometer), sun photometer and reanalysis data (Yang et al., 1999;Li et al., 2003;Prasad and Singh, 2009;Kwon et al., 2010). It can be also used for



evaluating the improvements of numerical weather prediction (NWP) using GPS-PWV (Gutman and Benjamin, 2001;Song et al., 2004). The time series of GPS-PWV over a GPS station have been used to study the temporal variation of PWV such as seasonal and diurnal variation patterns over the site of the station. GPS-PWV has been used to investigate the spatial variation in PWV over the region covered by the stations (Champollion et al., 2004;Jin and Luo, 2009;Van Baelen and

Penide, 2009).

The GPS-derived ZTD, i.e. GPS-ZTD, over a GPS station can be expressed as the sum of the zenith hydrostatic delay (ZHD) and zenith wet delay (ZWD) (Saastamoinen, 1972). The ZWD mainly stems from WV in the atmosphere below 10 km height. It can be converted to PWV by multiplying a dimensionless conversion factor that is a function of atmospheric-weighted-mean temperature ($T_m$), as expressed below (Askne and Nordius, 1987;Davis et al., 1985;Jade et al., 2005).

$$PWV = \Pi \cdot (ZTD - ZHD) = \Pi \cdot ZWD \tag{1}$$

$$\Pi = \frac{10^6}{\rho_w R_v \left(k_3 / T_m + k_2'\right)} \tag{2}$$

$$T_m = \frac{\int_h^{h_T} \rho_v \, dz}{\int_h^{h_T} \rho_v / T \, dz} \tag{3}$$

where, $\Pi$ is the conversion factor; $\rho_w$ and $\rho_v$ are the density of liquid water and WV respectively; $R_v$ is the specific gas constant for the air; $k_2'$ and $k_3$ are the atmospheric refractivity constants given in Bevis et al. (1994); $e$ is the WV pressure

(in hPa); and $T$ is the absolute temperature of the atmosphere (in Kelvin (K)).

In Eq. (3), $h_T$ is the height of the top of the troposphere and $h$ is the height of the GPS station. The reason for integrating from $h$ to $h_T$ is that WV only exists within the troposphere. It is also noted that in Eq. (1), both PWV and ZWD are in the unit of millimetres.

To determine $T_m$ over a GPS station or at any given point by Eq. (3), the profiles of atmospheric temperature and WV

pressure over the point are required, but they are very difficult to be obtained. Hence, the following three methods are often used: (1) ray tracing, (2) regression model ($T_m = a + b \cdot T_s$, i.e. the Bevis relationship between $T_m$ and atmospheric temperature $T_s$) and (3) empirical model. Each of these methods is explained below (Bevis et al., 1994;Ross and Rosenfeld, 1997;Ross and Rosenfeld, 1999).

$T_m$ derived from the ray tracing method is through an integral from radiosonde or NWP model data. In practice, this method

is rarely used due to its low temporal resolution nature and unavailability in real-time/near real-time (RT/NRT) (Wang et al., 2016). As for the regression model $T_m = a + b \cdot T_s$, the coefficients ($a$ and $b$) for different areas and seasons are determined/estimated from meteorological measurements by the least squares (Wang et al., 2011;Bevis et al., 1992;Schueler et al., 2001;Mendes et al., 2000;Emardson and Derks, 2000). The root-mean-square error (RMS) of $T_m$ from the regression model is in the range of 2−5 K. However, the primary limitation of the regression model is lack of temperature




measurements at most GPS stations. Thus an empirical $T_m$ model is used as a practical alternative for GPS meteorology. Although the accuracy of empirical models is lower than that of aforementioned methods, it can be used to calculate the $T_m$ in real-time/near real-time (RT/NRT) since only coordinates of the site and time are required. Table 1 summarises popular empirical $T_m$ models adopted by researchers in the last three years. The Data Source column presents the type and time span of the data used to develop the model, e.g., NCEP-DOE Atmospheric Model Intercomparison 2 (NCEP2) data, ERA-Interim data released by the European Centre for Medium-range Weather Forecasts (ECMWF) and the Global Geodetic Observing System (GGOS) data generated from ECMWF reanalysis data.

Inspired by the way how the global pressure and temperature (GPT) model is developed (Böhm et al., 2007), Yao et al. (2012) developed the season-specific Global Weighted Mean Temperature (GWMT) model based on radiosonde data of 135 global stations in the period 2005−2009. Its RMS accuracy of $T_m$ over the ground was shown to be around 4.6 K. Due to its poor performance in the southern Pacific Ocean, the coefficients were recalculated for an updated model — GTm-II using ocean $T_m$ calculated from the GPT model and the Bevis $T_m−T_s$ relationship (Yao et al., 2013). This GTm−II model was further improved into GTm−III using GGOS surface $T_m$ by taking semi-annual and diurnal variations in $T_m$ into account (Yao et al., 2014a). Chen et al. (2014) expressed the nonlinear model in GTm−III into a linear model based on the trigonometric function conversions and developed it further into GTm_N. Unlike the spherical harmonics applied in GTm_N, Chen and Yao (2015) established GTm-X based on the semi-annual and diurnal variations in $T_m$ with a global resolution of a 1°×1° geographical grid. More details for these three models can be found in Appendix B. It is worth noting that UNB3m and GPT2w are not specific $T_m$ models even though they can be used to derive $T_m$ (Leandro et al., 2008;Bohm et al., 2015).

However, diurnal variation and lapse rate of $T_m$ are either ill-modelled or ignored in most of these empirical models mentioned above. Therefore, this study presents our recent development towards an improved $T_m$ model (i.e. GWMT–D). This paper is structured as follows. Section 2 describes the data sets and the integral method for obtaining $T_m$, followed by the methodology of using global NCEP2 data in the four-year period 2010−2013 to develop the new model in Section 3. The performance of the new model GWMT−D is assessed in Section 4 through comprehensive comparisons against three other selected models using reference $T_m$ derived from NCEP2, radiosonde and GGOS data in 2014. Conclusions are presented in Section 5.

## 2. Data for the determination of $T_m$

Three data sets used to calculate $T_m$ include the NCEP2, radiosonde, and GGOS data with various temporal and spatial resolutions. The first data set – NCEP2 data in the period 2010−2013 is used to develop the new GWMT−D model, while all these three data sets in 2014 are used to evaluate GWMT−D as well as the other three selected empirical $T_m$ models.



## 2.1 NCEP2 data

The monitoring of global climate changes is the main aim of the National Centers for Environmental Prediction/National Center for Atmospheric Research (NCEP/NCAR) reanalysis data. A state-of-the-art analysis and forecast system has been used to assimilate multi-source data since 1948 and the American NCEP2 data set is an update version to its former reanalysis data (available on www.esrl.noaa.gov/psd/data/gridded/data. ncep.reanalysis2.pressure.html) (Kanamitsu et al., 2002).

The NCEP2 data set has a vertical resolution of 17 pressure levels ranging from 1000 to 10 hPa, a horizontal resolution of 2.5 °×2.5 ° and a temporal resolution of six hours (namely, at 0, 6, 12, 18 UTC), respectively. The data are organised in full 360 ° latitude circles beginning at 90 °N and stepping southward to 90 °S. In this study, temperature, geopotential height, pressure and humidity included in the pressure-level data over the period of 2010−2014 are selected for the development and validation of the new GWMT−D model.

## 2.2 Radiosonde data

Radiosonde profile data from 585 Integrated Global Radiosonde Archive (IGRA) stations over the globe in 2014 (Figure 1) are selected to validate the new GWMT−D model. They are retrieved from the upper-air archive at the website of University of Wyoming (available on http://weather.uwyo.edu/upperair/sounding.html). The daily observations at a site usually consist of 1−4 radiosonde observations, containing pressure, temperature, geopotential height, dew point depression, relative humidity, and mixing ratio at the surface, tropopause, and standard pressure levels (i.e., 1000, 925, 850, 700, 600, 500, 400, 300, 250, 200, 150, 100, 70, 50, 30, 20, and 10 hPa) (Wang et al., 2005). $T_m$ values are obtained through numerical integration (see Appendix A) under the assumption that the collected pressure, temperature and humidity measurements are along the zenith direction, even though radiosonde balloons often drift away from the vertical direction, especially in windy days.

In addition, raw radiosonde measurements are regarded as outliers and rejected under the following conditions:

(1) the height of the first record in the profile is larger than 20 m above the ground;

(2) the difference of heights between two successive pressure levels is larger than 10 km;

(3) the gap between two successive atmospheric pressure levels is larger than 200 hPa;

(4) the total number of valid radiosonde levels is less than 20;

(5) the highest humidity level is far lower than the height of the top troposphere obtained from an empirical model (200~350 hPa) (Liu, 2015);

(6) the height of the last data record in the profile is lower than 20 km.



### 2.3 Surface $T_m$ from GGOS Atmosphere

In this study, global surface $T_m$ values (i.e. the lower limit of the integral boundary in Eq. (3) is the surface of the site) are used for the validation of the new GWMT−D model and the three selected empirical $T_m$ models. GGOS Atmosphere publishes the daily global surface $T_m$ with a horizontal resolution of 2 °×2.5 °(latitude and longitude) for 00, 06, 12 and 18 UTC. Due to the fact that the GGOS data set has been applied in the development GTm−III, the surface $T_m$ from the GGOS data set is also used in the performance assessment of three selected empirical $T_m$ models. Nevertheless, the discrepancies between these different data sets are noticeable and may affect the validation results, which will be shown in Section 4.

### 3. GWMT−D model

The NCEP2 data from the four-year period 2010−2013 are employed to develop the new GWMT−D (D stands for diurnal variation) model. The detailed procedure for the calculation of $T_m$ from NCEP2 data using temperature, geopotential height, and relative humidity profile is outlined in Appendix A. Note that geopotential height in the radiosonde and NCEP2 data needs to be converted to ellipsoidal height (refer to Appendix A), which is simplified as 'height' hereafter.

### 3.1 Improvements in GWMT−D

Compared with other empirical $T_m$ models, the improvement achieved by the new GWMT−D model is the modelling of the diurnal variation and lapse rate in $T_m$. The $T_m$ lapse rate in this paper is the decreasing rate in $T_m$ (Bevis et al., 1994;Yao et al., 2012). NCEP2-derived $T_m$ values are for 17 pressure levels and the heights of these pressure levels are dynamic. In order to investigate a time series of $T_m$ at fixed heights over a site, NCEP2-derived $T_m$ are interpolated for four selected heights — 0, 2, 5 and 9 km using the spline interpolation method to avoid the Runge's phenomenon (Fornberg and Zuev, 2007). For the GWMT−D model, the $T_m$ time series at each of the reference times (00, 06, 12, 18 UTC) of day are assumed to follow a seasonal cycle. It can be expressed by a function of day of year (*DOY*):

$$T_m(DOY) = \alpha_1 + \alpha_2 \cos(2\pi \frac{DOY}{365.25}) + \alpha_3 \sin(2\pi \frac{DOY}{365.25}) + \alpha_4 \cos(4\pi \frac{DOY}{365.25}) + \alpha_5 \sin(4\pi \frac{DOY}{365.25}) \tag{4}$$

where, $\alpha_1$ is the yearly mean value; $\alpha_2$ and $\alpha_3$ are the coefficients of the annual variation; $\alpha_4$ and $\alpha_5$ are that of the semi-annual variation.

These coefficients are estimated using the least squares method and the observations are a time series of $T_m$ values at the specific reference time over the site. The voxel-based feature of the model's coefficients is where this new model primarily differs from all the others. The new model is a four-dimensional (4D) global $T_m$ field with a re-sampled horizontal resolution of 5 °×5 °at the four vertical levels and the four reference times.



### 3.1.1 Diurnal variation

Annual and semi-annual variations in a $T_m$ time series over a site obtained from NCEP can be detected using the spectrum analysis (Chen et al., 2014). Although simple sine and cosine functions have been widely used to model the diurnal variations of $T_m$, little study has been conducted to analyse the periodic nature of the diurnal variation in $T_m$. Diurnal

variations in different seasons and locations are first investigated to study the voxel-based modelling process in GWMT−D. Figure 2 shows three examples of the diurnal variation at 2 km above the ground for different latitudes. It clearly shows that the diurnal variation in $T_m$ cannot be modelled by simple trigonometric functions like what is used by GTm–III model (see Appendix B), since it is different across different seasons and locations. This paper takes this feature into account and a new modelling procedure is designed to capture the diurnal variation, i.e. $T_m$ values at any other time are obtained by a spline

interpolation method.

### 3.1.2 Vertical lapse rate of $T_m$

The $T_m$ lapse rate along the vertical direction can be affected by several factors, e.g., the moisture content of air, atmospheric pressure and the surface height. Figure 3 illustrates the global distribution of annual mean $T_m$ lapse rate in the height layer from the ground up to 2 km in 2013. It shows that global annual mean $T_m$ lapse rate varies with latitude and land-sea

distribution is around −4.5 K/km. The result is similar with what has been found by other recent studies (Chen et al., 2014;Yao et al., 2015). Therefore, it is essential to consider the vertical variation of $T_m$ with locations instead of using a constant on a global scale in order to build a more accurate empirical $T_m$ model (see Section 4.3).

Based on an analysis to the voxel-based modelling of diurnal variation in $T_m$, four specific height levels (0, 2, 5 and 9 km) based on a piecewise linear interpolation algorithm are selected covering most of the troposphere in the new GWMD−D

model. All global $T_m$ values from these heights for each reference time are then calculated (see Eq. (3)). The $T_m$ value for any other heights can be obtained by interpolating its two nearest height levels. This improvement is a distinguished feature of the new model in comparison with the aforementioned empirical models where a constant $T_m$ lapse rate is adopted for the different heights over the globe.

### 3.1.3 Data span used in $T_m$ modelling

Another important task is to determine the optimal length of reanalysis data required for the development of empirical $T_m$ models. Long-term $T_m$ time series over the globe can be used for climatological analysis, but its temporal correlation may be too weak to be considered in the $T_m$ modelling process. This suggests that a short period of data may lead to an unreliable result. Consequently, an optimal length of period needs to be investigated.

Different sets of coefficients of the GWMT−D are calculated using the NCEP2-derived $T_m$ data for a period of one (2013) to

nine years (2005−2013). The model-derived $T_m$ values from GWMT−D with these different sets of coefficients are compared with one-year NCEP2-derived $T_m$ time series (2014) at five pressure levels (1000, 925, 850, 700 and 600 hPa).



Table 1 lists the statistical results of the comparison. In this research, the NCEP2 $T_m$ time series from the four-year period are adopted to develop the GWMT−D model for its best fitting results (shown in bold fonts).

## 3.2 The procedure to determine $T_m$ using GWMT−D

Assuming $T_m$ at the target location $(\varphi, \lambda, h)$ on day $(DOY)$ and hour $(HOD)$ is $T_m (\varphi, \lambda, h, DOY, HOD)$, the key steps of determining $T_m$ can be described as follows:

(1) Determining two surfaces at the two reference height levels closest to $h$, (see Figure 4 in grey) and the other four vertical surfaces containing the eight voxels closest to $(\varphi, \lambda)$, then calculating the $T_m$ on the eight voxels using the equation below

$$T_m(\varphi_i, \lambda_j, h_l, t_k) = \alpha_1(\varphi_i, \lambda_j, h_l, t_k) + \alpha_2(\varphi_i, \lambda_j, h_l, t_k)\cos(2\pi\frac{DOY}{365.25}) + \alpha_3(\varphi_i, \lambda_j, h_l, t_k)\sin(2\pi\frac{DOY}{365.25})$$
$$+ \alpha_4(\varphi_i, \lambda_j, h_l, t_k)\cos(4\pi\frac{DOY}{365.25}) + \alpha_5(\varphi_i, \lambda_j, h_l, t_k)\sin(4\pi\frac{DOY}{365.25})$$

(5)

where $\varphi_i$ and $\lambda_j$ are the latitude and longitude of the vertex (at a 5 °×5 °resolution); $l$ ($l = 1, 2, 3, 4$) is the index of the reference height $h_l$ corresponding to 0, 2, 5 or 9 km respectively; $t_k$ ($k = 1, 2, ...5$) is the index of the reference time corresponding to 0, 6, 12, 18 and 24 UTC respectively.

(2) Performing a vertical linear 1D interpolation for the point at the height of $h$ using the $T_m$ values of the two voxels in each of the four vertical edges (see the four corners in the dashed rectangular in Figure 4)

$$T_m(\varphi_i, \lambda_j, h) = T_m(\varphi_i, \lambda_j, h_l) + \frac{T_m(\varphi_i, \lambda_j, h_{l+1}) - T_m(\varphi_i, \lambda_j, h_l)}{h_{l+1} - h_l} \cdot (h - h_l)$$

(6)

(3) Performing a horizontal bilinear 2D interpolation using the $T_m$ values of the four corners in the dashed rectangular to obtain the target point's $T_m$ by

$$p = (\lambda - \lambda_j)/5, \quad q = (\varphi - \varphi_i)/5$$

(7)

$$T_m(\varphi, \lambda, h) = (1-p)(1-q) \cdot T_m(\varphi_i, \lambda_j, h) + p(1-q) \cdot T_m(\varphi_i, \lambda_{j+1}, h)$$
$$+ (1-p)q \cdot T_m(\varphi_{i+1}, \lambda_j, h) + pq \cdot T_m(\varphi_{i+1}, \lambda_{j+1}, h)$$

(8)

All the notations in Eq. (7) and Eq. (8) can be found in the dashed rectangular in Figure 4. The number '5' in Eq. (7) is the horizontal resolution of the new model.

(4) After the above spatial interpolations are performed, the final step is a spline interpolation in the time domain from four reference times (i.e. from 0, 6, 12, 18, and 24 UTC) of the day closest to $t_k$.

## 4. Validation of $T_m$ models

Different empirical $T_m$ models (Table 1) are developed based on different data sets. The accuracies of these models claimed in relevant literatures are referenced to different reference values (e.g., He et al., 2013;Chen et al., 2014;Yao et al., 2014a).



Consequently, cross comparisons of these accuracy values for their performance may not be appropriate. In this study, the performance of three selected empirical $T_m$ models and the new GWMT−D model are assessed using the same reference $T_m$ values derived from NCEP2, GGOS and radiosonde data sets.

Due to the fact that GTm_X is unavailable to the public and GWMT and GTm-II have been proven inferior to GTm−III,

GWMT−IV and GTm_N, only the GTm−III, GWMT−IV, GTm_N models and our new GWMT−D model are assessed. The methodologies for obtaining $T_m$ from NCEP2, radiosonde data sets are given in Appendix A. The two statistical quantities used to measure the performance of these models are bias and RMS, which are calculated by

$$Bias = \frac{1}{N}\sum_{i=1}^{N}\left(T_m^{Ci} - T_m^i\right) \tag{9}$$

$$RMS = \sqrt{\frac{1}{N}\sum_{i=1}^{N}(T_m^{Ci} - T_m^i)^2} \tag{10}$$

where, $T_m^C$ and $T_m$ are the $T_m$ values from the models and the reference respectively, and $N$ is the number of the samples.

## 4.1 Comparison with NCEP2 data

Section 3.1.2 shows that the piecewise linear algorithm for vertical $T_m$ interpolation in GWMT−D is better than the direct modelling of $T_m$ lapse rate in GWMT−IV or the constant-value method used in both GTm−III and GTm_N. Particularly, the constant-value method performs poorly in both temporal and spatial domains.

Although more than 80% of the International GNSS Service stations and IGRA stations used in this study have a station altitude below 1 km, the highest height of the IGRA stations selected for the comparisons can reach up to 5 km. As a result, only the statistics of all global grid points on two pressure levels 925 hPa (~0.6 km) and 600 hPa (~5 km) are given in Table 3. Nevertheless, similar results can be obtained from the new GWMT−D model on the other pressure levels less than 600 hPa (refer to Table 2). One can find from Table 3 that GWMT−D significantly outperforms all the other three empirical

models.

Figures 5−6 illustrate the distribution of the RMS (not the mean RMS of all grid points listed in Table 3) of the differences between the $T_m$ derived from the models and the NCEP2 data in 2014 on two pressure levels. Figures 5(d) and 6(d) present the best agreement for GWMT−D over the globe. On the pressure level of 925 hPa, more than 91% of the grid points had RMS less than 5 K, compared with 77% from GTm_N and GTm−III, and less than 71% from GWMT−IV. Whilst on the 600

hPa level, GWMT−IV is worse, especially in the Arctic Circle. The RMS values of GWMT−D ranged from 1.27 K to 11.55 K, outperforming the other three models with a global average RMS of only 4.73 K and an approximately 25% improvement over the other models.

It is worth pointing out that all these four models have relatively low RMS values near the tropical areas, and all have a similar performance globally except for the Antarctic. This finding is consistent with recent studies, (e.g., He et al. (2013),

Chen et al. (2014) and Yao et al. (2014a)). It may be explained by the fact that the $T_m$ on this pressure level is not directly





derived from actual measurements since the terrain of the Antarctic is generally higher than the pressure level of 1000 hPa. In other word, the extrapolated $T_m$ on this pressure level over this area may contain large systematic biases.

## 4.2 Comparison with GGOS data

The GGOS surface grid $T_m$ data in 2014 is used as the reference in this section to evaluate the performance of the four

models. The statistical results of the same four selected models are shown in Table 4 and Figure 7 for their global distribution. The GTm−III performed the best this time because the GGOS Atmosphere data derived from ECMWF reanalysis data are used in the development of the GTm−III. From Table 3, GWMT−D is almost unbiased while the GTm−III showed a bias of −1.25~−1.31 K in comparison with the NCEP2-based $T_m$. In contrast, a bias of about +1.2 K (warmer) compared to the GGOS-based $T_m$ is found with GWMT−D (see Table 4). This discrepancy of 1.2 K between the

NCEP2-derived and GGOS-derived $T_m$ may result from differences of NWP systems, e.g. different observations, physical models, data assimilation processes and boundary conditions (Buizza et al., 2005).

Nevertheless, the good performance of GWMT−D indicates that the modelling process of $T_m$ can significantly improve the model's accuracy. Figure 7d indicates that GWMT−D has RMS values of less than 6 K at most grids, except the areas in the Antarctic, northeast North America and Middle East (6−10 K).

**4.3 Comparison with radiosonde data**

These four empirical $T_m$ models of interest are also evaluated using independent measurements (i.e. radiosonde). A number of comparisons are carried out in this section, including:

      (1) Surface $T_m$ values calculated from radiosonde measurements are used as the reference to compare with various model-derived surface $T_m$;

20       (2) $T_m$ derived from the GWMT-D and three other selected empirical models is compared with radiosonde-derived $T_m$ to investigate models' performance in different heights;

      (3) The accuracy of the $T_m$ models in different seasons is also investigated.

Figure 8 illustrates the RMS of model-derived surface $T_m$ in 2014 at the 585 selected radiosonde stations. The spatial (horizontal) variation in the accuracy of these models can be seen from this figure. An accuracy of better than 8 K has been

achieved at most stations for the GWMT−D (Figure 8d) and a similar accuracy can be achieved by the GTm_N as well (Figure 8c). These two models outperformed the other two models: GTm−III and GWMT−IV, especially in the Middle East, the Siberia and the South Africa regions.

Figure 9 shows the histogram of the difference (i.e. model-derived $T_m$ minus radiosonde-derived $T_m$) at all heights from 0 to 9 km in terms of the mean, standard deviation, median and mode values. One of the new trials in this research is that we use

mode and median values to estimate the sample bias. The main advantage of using the mode and median values is more robust than arithmetic mean, especially, in skewed distributions (see Figure 9b and 9c). As a result, a warm (cold) bias of 3.8





K (−4.4 K) can be found in the GTm_N (GWMT−IV). The histograms of both GTm−III and GWMT−D (Figure 9a and 9d) are normally distributed and the GWMT−D is slightly better than the GTm−III.

The entire radiosonde-derived $T_m$ is grouped into three height intervals 0−2, 2−5, and 5−9 km, according to their station heights. The results are listed in Table 5 for the accuracy comparison between the GWMT−D and other models in different height intervals. It can be concluded that the accuracies of all the models except for the GWMT−D are significantly degraded with the increase of the height of the site. In contrast, it shows that the accuracy of GWMT−D is stable in three different height ranges. Comparing with the GTm_N model, better performance of the GTm−III may result from the discrepancy between GGOS surface $T_m$ data (ECMWF reanalysis data) used by GTm-III and NCEP reanalysis data used by GTm_N.

The RMS values of GWMT−D, GTm−III and GTm_N are plotted in the Figure 10 as a function of height relative to the ground surface in order to reveal the representative effect of terrain on the models. The GWMT−D model's RMSs are all in the range of 4−5 K, while the other two models' RMS values have much larger values and increase rapidly with the increase of height. It is noted that the GWMT−IV model is excluded due to its poorer performance shown in Table 5. The GWMT−D's result suggests its accuracy is better than 5 K, even at the top of the troposphere.

Figure 11 shows the monthly-mean RMSs of these four model for comparison of monthly or seasonal performance of these models. We can see that the monthly-mean RMSs of all the models vary with month (or season) and only the GTm_N shows a variation pattern opposite to that of the other three models. The GWMT−D and GWMT−III give very similar results in both pattern of variation and monthly-mean RMSs. The GWMT−IV performs the worst and GWMT−D performs the best, among all these four models.

### 4.4 Impact of $T_m$ on GPS-derived PWV

The purpose of determining $T_m$ is to convert ZWD of GPS signals to PWV for the case that meteorological measurements are not available. GPS measurements are not used here in order to remove errors in the determination of both ZWD and PWV that in the refractivity constant since the $T_m$ is the main focus of this study. Using Equation (1), the relationship of the RMSs between $T_m$ and PWV is given by

$$\frac{RMS_{PWV}}{PWV} = \frac{RMS_{\Pi}}{\Pi} = \frac{k_3 \cdot RMS_{T_m}}{(k_3/T_m + k_2')T_m^2} = \frac{k_3}{(k_3/T_m + k_2')T_m} \cdot \frac{RMS_{T_m}}{T_m} \tag{11}$$

where, the three $RMS$s are defined for the differences between observed and true values (more details see Appendix C) and the relative error of PWV can be defined as $RMS_{PWV}/PWV$ here.

Figure 12 illustrates the global distribution of both $RMS_{PWV}$ and $RMS_{PWV}/PWV$ obtained from Equation (11) and radiosonde data in 2014. The value of $RMS_{T_m}$ used here is obtained from section 4.3. $PWV$ and $T_m$ are set to annual mean values. Some radiosonde stations have been removed with insufficient observations or near the polar areas. As we can see that the global mean values of $RMS_{PWV}$ and $RMS_{PWV}/PWV$ are around 0.25 mm and 1.3%, respectively.



## 5. Conclusion

$T_m$ is a critical parameter in PWV determination using the GPS atmospheric sounding technique. Robust $T_m$ models are required as a practical alternative of the conventional methods such as the ray tracing and regression methods, if in situ meteorological measurements cannot be obtained in RT/NRT. The variations of global lapse rate and diurnal fluctuations can significantly affect the accuracy of the $T_m$ determination since these variations are either ill-modelled or ignored in the exiting empirical $T_m$ models. Furthermore, no comprehensive inter-comparison has been carried out among empirical $T_m$ models with the same reference $T_m$. Therefore, a new voxel-based $T_m$ model, namely GWMT−D, has been developed in this study using global NCEP2 data from 2010 to 2013. This newly developed model takes advantage of voxel-based modelling method to effectively capture the diurnal variations and lapse rate in $T_m$. The new model is compared with three selected models including GTm−III, GWMT−IV and GTm_N using the NCEP2, GGOS surface $T_m$, and radiosonde data sets in 2014 as the reference.

It is shown that GWMT–D is unbiased and can achieve a RMS accuracy of 4~5 K for different seasons and locations for NCEP2 and radiosonde data sets, with an improvement of around 25% over the other three models. The comparisons with GGOS surface $T_m$ data have shown that GWMT–D is slightly worse than that of GTm–III with a bias of ~1.2 K, due to the difference between NCEP2 and ECMWF reanalysis data. This bias is not negligible, especially for the Antarctic. It is also suggested that the coefficient sets of empirical $T_m$ models (e.g., GWMT–D) need to be re-determined regularly using state-of-the-art data source. The new GWMT−D model can provide an alternative $T_m$ determination method to RT/NRT PWV remote sensing system so that continuous operation of this system can be maintained when in-situ meteorological measurements are unavailable. Around 1.3% relative error or 0.3 mm RMS in PWV will result from the new $T_m$ model for ground stations. Due to the fact that radiosonde measurements are mainly taken on the land, further inter-comparisons between empirical $T_m$ models and other measurements over the ocean need to be investigated, e.g., Constellation Observation System of Meteorology, Ionosphere, and Climate (COSMIC).

## Appendix A: Determination of $T_m$ and water vapour pressure

This appendix presents the method of $T_m$ and water vapour pressure determination. If layered meteorological data (e.g., Reanalysis and radiosonde) are available, the numerical integration in the Eq. (3) can be approximated as with:

$$T_m \approx \frac{\sum_{i=1}^{n}\left(\dfrac{e_i}{T_i} + \dfrac{e_{i+1}}{T_{i+1}}\right)\dfrac{\Delta z_i}{2}}{\sum_{i=1}^{n}\left(\dfrac{e_i}{T_i^2} + \dfrac{e_{i+1}}{T_{i+1}^2}\right)\dfrac{\Delta z_i}{2}} \tag{A1}$$

where, $e_i$ and $T_i$, $e_{i+1}$ and $T_{i+1}$ are the water vapour pressure, temperature respectively on the lower and upper boundary of the $i$th layer of the atmosphere; $\Delta z_i$ is the thickness of the $i$th layer; and $n$ is the total number of layers.




It should be noted that the height used in NCEP2 and radiosonde data is a geopotential height, which is widely used in meteorology, whilst the height used in the Eq. (A1) is a geometric height. The equations for the conversion of a geopotential height to a geometric height (ellipsoidal height) are (Aparicio et al., 2009):

$$h = \frac{R_e(\varphi) \cdot H}{\frac{g(\varphi)}{g_0} R_e(\varphi) - H} \tag{A2}$$

$$g(\varphi) = 9.80620(1 - 2.6442 \times 10^{-3} \cos 2\varphi + 5.8 \times 10^{-6} \cos^2 2\varphi) \tag{A3}$$

$$R_e(\varphi) = \frac{a}{1 + f + m - 2f \sin^2 \varphi} \tag{A4}$$

where, $\varphi$ is the latitude, $h$ is the geometric height (in km) and $H$ is the geopotential height (in km); the constant $g_0$ is assigned to 9.80665 m/s²; $g(\varphi)$ is the gravity acceleration along the plumbline; $R_e(\varphi)$ is the radius of curvature of the Earth at the latitude of $\varphi$; and the parameters $a = 6378.137 \ km, f = 1/298.257223563, m = 0.00344978650684$.

Since the humidity in layered meteorological data is recorded as dew point temperature ($T_d$) or relative humidity ($RH$) or specific humidity ($q$) instead of partial pressure of water vapour ($e$). The water vapour pressure needs to be computed first in the determination of $T_m$ with $T_d$, $RH$ and $q$, i.e.

$$e = f \cdot 6.112 \exp\left(\frac{17.62 \, t}{243.12 + t}\right) \tag{A5}$$

$$e = \frac{qP}{q + \varepsilon(1 - q)} \tag{A6}$$

$$\log_{10}(e)_{liquid} = \log_{10}\left(\frac{RH}{100}\right) + \log_{10}(f) + 10.79574\left(1 - \frac{273.16}{T}\right) - 5.028 \log_{10}\left(\frac{T}{273.16}\right)$$
$$+ 1.50475 \times 10^{-4}\left(1 - 10^{-8.2969\left(\frac{T}{273.16} - 1\right)}\right) + 0.42873 \times 10^{-3}\left(10^{-4.76955\left(1 - \frac{273.16}{T}\right)} - 1\right) + 0.78614 \tag{A7}$$

$$\log_{10}(e)_{ice} = \log_{10}\left(\frac{RH}{100}\right) + \log_{10}(f) - 9.096853\left(\frac{273.16}{T} - 1\right)$$
$$- 3.566506 \log_{10}\left(\frac{273.16}{T}\right) + 0.876812\left(1 - \frac{T}{273.16}\right) + 0.78614 \tag{A8}$$

where t is the temperature in Celsius degree, and t = T − 273.15 ; $\varepsilon$ = Mw/Md is the ratio of the molar masses of vapour and dry air, respectively; f(P) is enhancement factor defined as the ratio of the saturation vapour pressure of moist air to that of pure water vapour (WMO, 2000); Eq. (A7) and Eq. (A8) are deduced from the Goff's formulation and its units of water

vapour pressure are Pa (Goff, 1957). $T_m$ in this study is computed with relative humidity data. Note that interpolation of meteorological measurements is not applied in Eq. (A1).





## Appendix B: Empirical $T_m$ models

### B 1. UNB3m

Strictly, the UNB3m model is not a specific $T_m$ model, but it can be used to calculate $T_m$ from the following equation (Leandro et al., 2008):

$$T_m^{UNB3m} = (T_0 - \beta_T \cdot h)\left(1 - \frac{\beta_T R}{g_m(\lambda + 1)}\right) \tag{B1}$$

where, $T_0$ is the temperature at the mean sea level; $\lambda$ is the dimensionless water vapour pressure height factor; $\beta_T$ is temperature lapse rate; $g_m$ is the acceleration of gravity at the atmospheric column centroid; $R$ is the gas constant for dry air; $h$ is the height of unknown position.

The UNB3m model neglects the longitudinal variation in $T_m$. The meteorological variables in Eq. (B1), i.e., $T_0$, $\lambda$, and $\beta_T$, are linearly interpolated in latitudinal direction based on a simple look-up table.

### B 2. GPT2w

The GPT2w, an improved GPT model, was developed by Bohm et al. (2015). This empirical model can provide pressure, temperature, tropospheric delay as well as $T_m$ with the annual and semi-annual amplitudes. The updated model was established on a regular resolution of 5° with monthly meteorological data of 10 years (2001–2010) ERA-Interim. The GPT2w is not specifically designed for $T_m$ computation. The $T_m$ is calculated with Eq. (B5), but the coefficients in this equation are determined based on a regular grid of 5° or 1°.

### B 3. GWMT series models

The global weighted mean temperature (GWMT) series models are global models developed and consistently improved by Yao et al. using the state-of-the-art data sources and improved methodologies (Yao et al., 2015;Yao et al., 2014b;Yao et al., 2014a;Yao et al., 2013;Yao et al., 2012).

The GWMT model was based on spherical harmonics of degree nine and order nine and is a function of the geodetic coordinates of the site, as expressed below:

$$T_m^{GWMT} = \alpha_1 + \alpha_2 h + \alpha_3 \cos\left(\frac{DOY - 28}{365.25} 2\pi\right) \tag{B2}$$

$$\alpha_i = \sum_{n=0}^{9}\sum_{m=0}^{n} P_{nm}(\sin\varphi) \cdot \left[A_{nm}^i \cos(m\lambda) + B_{nm}^i \sin(m\lambda)\right] \quad (i = 1 \, or \, 3) \tag{B3}$$

where the globally mean lapse rate of $T_m$, $\alpha_2$, is −4.1 K/km; $\varphi$, $\lambda$ and $h$ are the latitude, longitude and height of the site respectively; $DOY$ is the day of year; $P_{nm}$ is the Legendre function; $A_{nm}^i$ and $B_{nm}^i$ in Eq. (6) are two coefficients estimated from the least-squares estimation.



The GTm−II model was identical to GWMT in theory but with different model coefficients.

Considering the semi-annual and diurnal variations in $T_m$, the GTm−III model can be expressed as:

$$T_m^{GTm-III} = \alpha_1 + \alpha_2 h + \alpha_3 \cos\left(\frac{DOY - C_1}{365.25} 2\pi\right) + \alpha_4 \cos\left(\frac{DOY - C_2}{365.25} 4\pi\right) + \alpha_5 \cos\left(\frac{HOD - C_3}{365.25} 2\pi\right)` \tag{B4}$$

where, $HOD$ is the hour of the day. The coefficients $\alpha_i$ ($i = 1, 2, ..., 3$) are expended to spherical harmonics similar with the
case in GWMT and GTm−II.

Since the adjustment model in Eq. (B4) for the GTm−III is non-linear, the coefficients determined may be unstable or biased. Chen et al. (2014) established the GTm_N model with a global grid of $2.5° \times 2.5°$ NCEP reanalysis data neglecting the diurnal variation in $T_m$. The GTm_N model linearizes the Eq. (B4) as (Chen et al., 2014):

$$T_m^{GTm-N} = \alpha_1 + \alpha_2 h + \alpha_3' \cos\left(\frac{DOY}{365.25} 2\pi\right) + \beta_3' \sin\left(\frac{DOY}{365.25} 2\pi\right) + \alpha_4' \cos\left(\frac{DOY}{365.25} 4\pi\right) + \beta_4' \sin\left(\frac{DOY}{365.25} 4\pi\right) \tag{B5}$$

All the aforementioned models are based on such an assumption that the vertical lapse rate of $T_m$ is the same over the globe, i.e. the $\alpha_2$ in these equations are constant scalars. In fact, this assumption is not always true (He et al., 2013). Therefore, the horizontal variation of $T_m$ lapse rate ($\beta$) is considered in the GWMT−IV model. It is a function of the horizontal location. Thus, the global $T_m$ at the height of $h$ can be expressed as a function of the mean sea level $T_m$ ($T_m^0$) and $\beta$ in Eq. (9), both of which can be further separated into annual and semi-annual components. Both amplitudes and initial phases parameters of annual and semi-annual variations are similarly expended into a spherical harmonics form.

$$T_m^{GWMT-IV}(h) = T_m^0 + \beta \cdot h \tag{B6}$$

**Appendix C: Approximated propagation of RMS**

Given a series of observations $V$ collected at the same time (Ning et al., 2016):

$$V_i = \tilde{V} + M + \varepsilon_i \tag{C1}$$

where, $M$ is a time-independent bias (systematic error); $\tilde{V}$ is the true value of observations $V_i$; and $\varepsilon_i$ is the zero-mean stationary Gaussian random error. Hence, the RMS of the difference between estimates and true values is given by

$$RMS_V = \sqrt{\frac{1}{N} \sum_{i=1}^{N} (V_i - \tilde{V})^2} = \sqrt{\frac{1}{N} \sum_{i=1}^{N} (M + \varepsilon_i)^2} \tag{C2}$$

where, $N$ is the total number of observations. Since the mean value of $\varepsilon_i$ will be close to zero for massive repeated observations, Equation (C2) can be approximately reduced to

$$RMS_V = \sqrt{\frac{1}{N} \sum_{i=1}^{N} (M^2 + \varepsilon_i^2)} = \sqrt{M^2 + \sigma_\varepsilon^2} \tag{C3}$$





where, $\sigma_\varepsilon$ is the standard deviation of $\varepsilon$. As can be seen from this equation, the RMS will be identical to standard deviation if observations are free of systematic bias. Consider a linear or nonlinear function $W = f(V): \mathbb{R} \mapsto \mathbb{R}$ whose RMS can be expressed by

$$RMS_W = \sqrt{\frac{1}{N}\sum_{i=1}^{N}[f(V_i) - f(\tilde{V})]^2} \tag{C3}$$

5  Using 1st order Taylor expansion, we have

$$f(V) - f(\tilde{V}) \approx (V - \tilde{V}) \cdot \left.\frac{\partial f(V)}{\partial V}\right|_{V=\tilde{V}} \tag{C4}$$

Substituting Equation (C4) into Equation (C3)

$$RMS_W \approx \left.\frac{\partial f(V)}{\partial V}\right|_{V=\tilde{V}} \cdot \sqrt{\frac{1}{N}\sum_{t=1}^{N}(V_i - \tilde{V})^2} = \left.\frac{\partial f(V)}{\partial V}\right|_{V=\tilde{V}} \cdot RMS_V \tag{C5}$$

As a result, the RMS of $f(V)$ can be approximately propagated from the that of $V$.

## Acknowledgements

We would like to thank the NOAA/OAR/ESRL PSD (Boulder, Colorado, USA) for providing NCEP-DOE Reanalysis 2 data (2010-2014), the Department of Atmospheric Science in the University of Wyoming for providing access to radiosonde data in 2014, the GGOS (Global Geodetic Observing System) Atmosphere for providing 2014 surface $T_m$ product. This research is partially supported by the Australian Research Council grant (ARC- LP130100243)

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





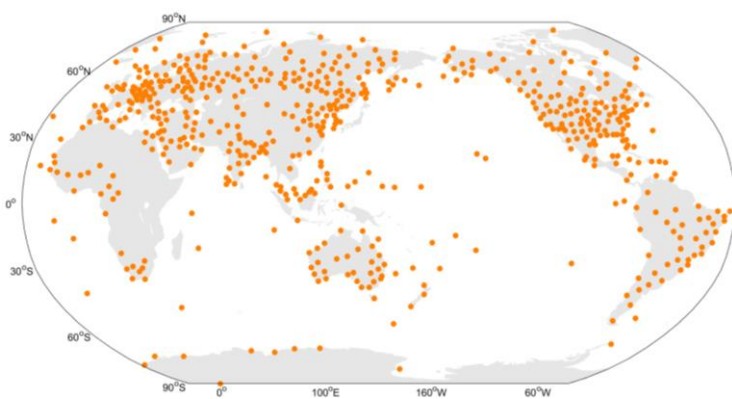

**Figure 1. Distribution of the 585 radiosonde stations selected to validate the new GWMT−D model (Only those data that pass a quality check are used).**

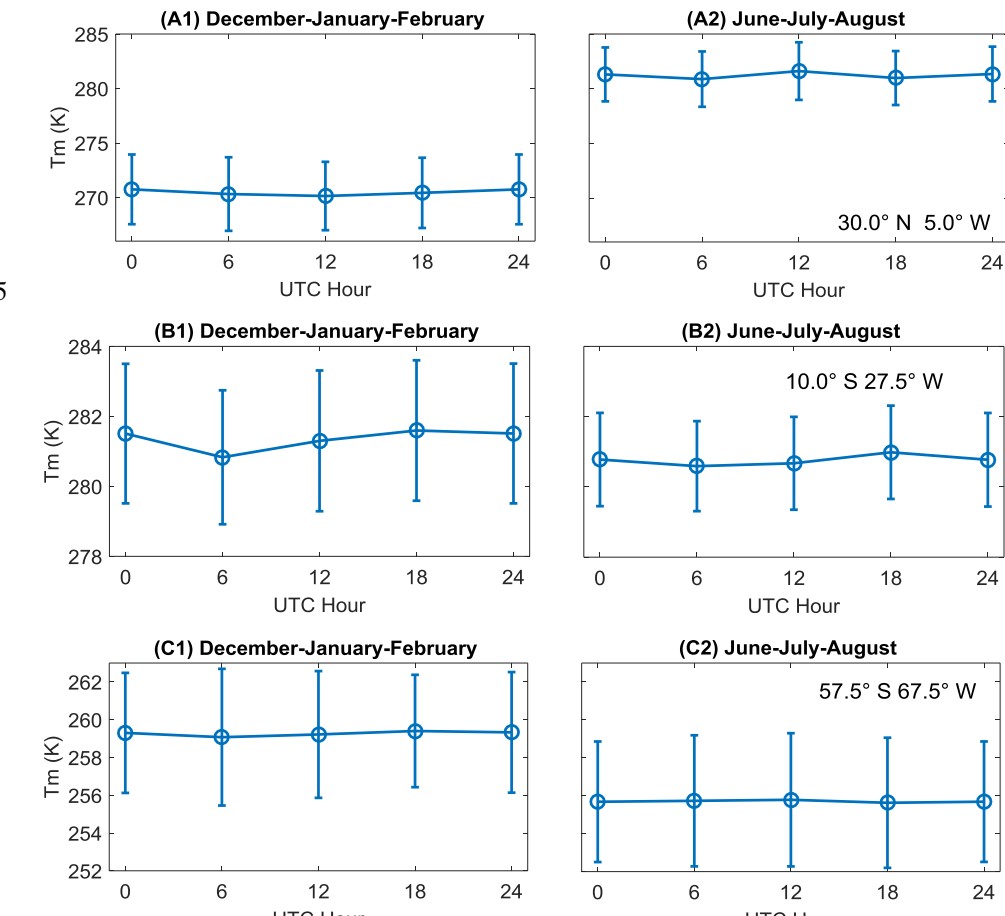

**Figure 2. Seasonal statistical results of $T_m$ (mean ± standard deviation) at 2 km height and four reference times during Dec-Jan-Feb and Jun-Jul-Aug in 2014.**





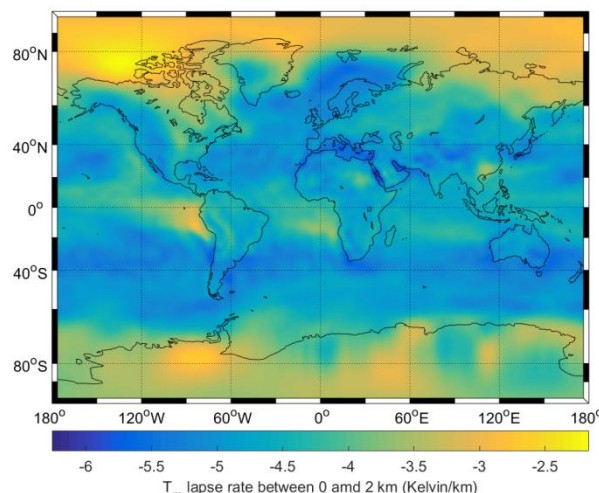

**Figure 3. Global annual mean $T_m$ lapse rate in the height interval 0–2 km from NCEP2 in 2013.**

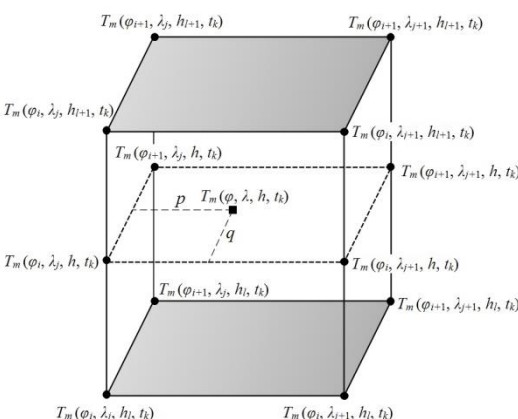

**Figure 4. Spatial interpolation for the target point located at $(\varphi, \lambda, h)$ using the $T_m$ values at the eight closest voxels determined by the GWMT−D model. The first interpolation is for each of the four vertical edges of the box, and the second interpolation is on the**
10 **2D plane at the height of the target point (the dashed rectangular).**





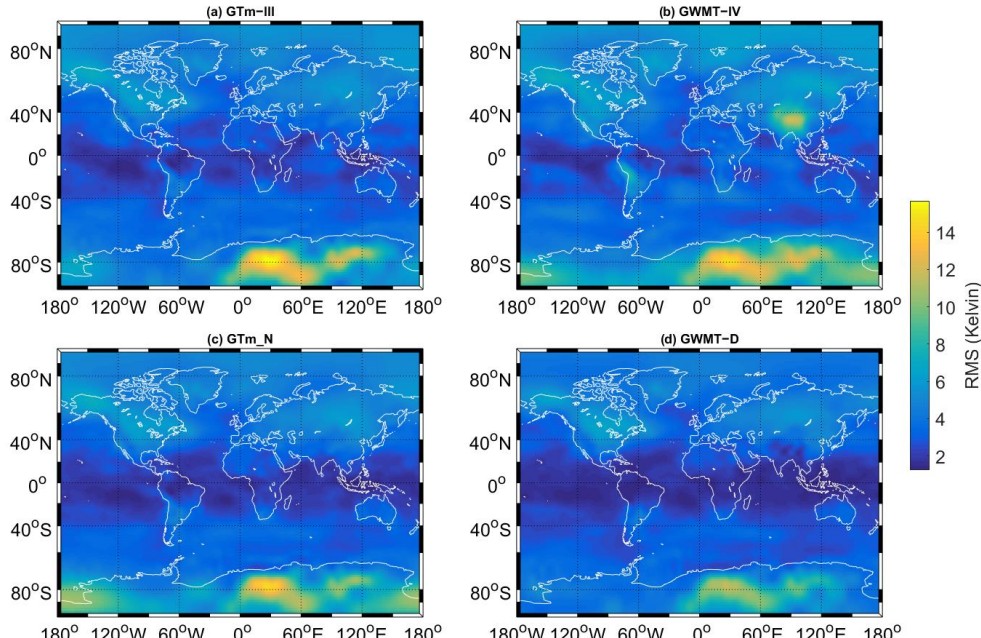

**Figure 5. The global RMS distribution of the differences between the $T_m$ derived from each model and the NCEP2 data on pressure level of 925 hPa in 2014.**

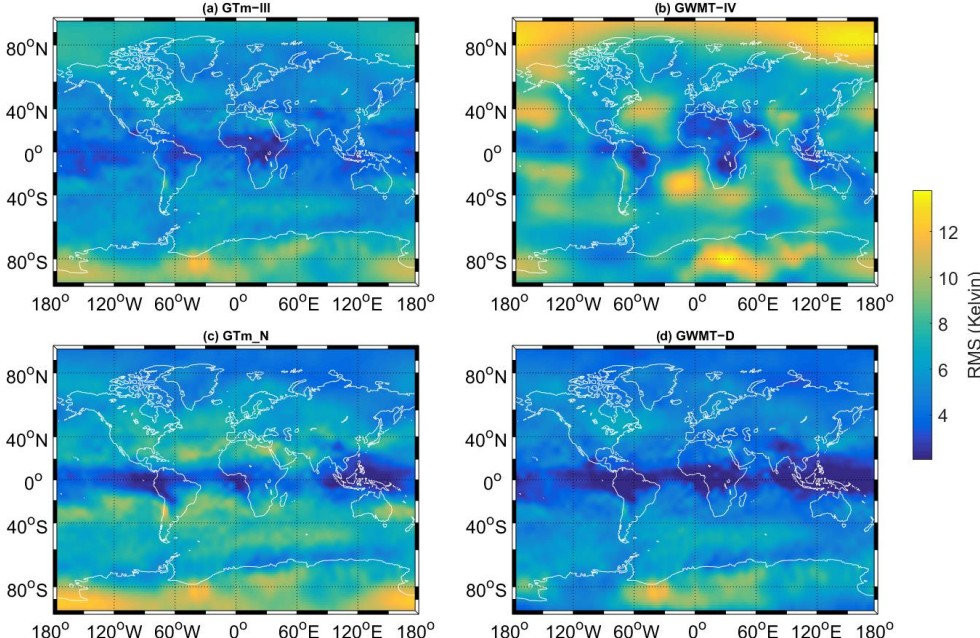

5 **Figure 6. The global RMS distribution of the differences between the $T_m$ derived from each model and the NCEP2 data on pressure level of 600 hPa in 2014.**





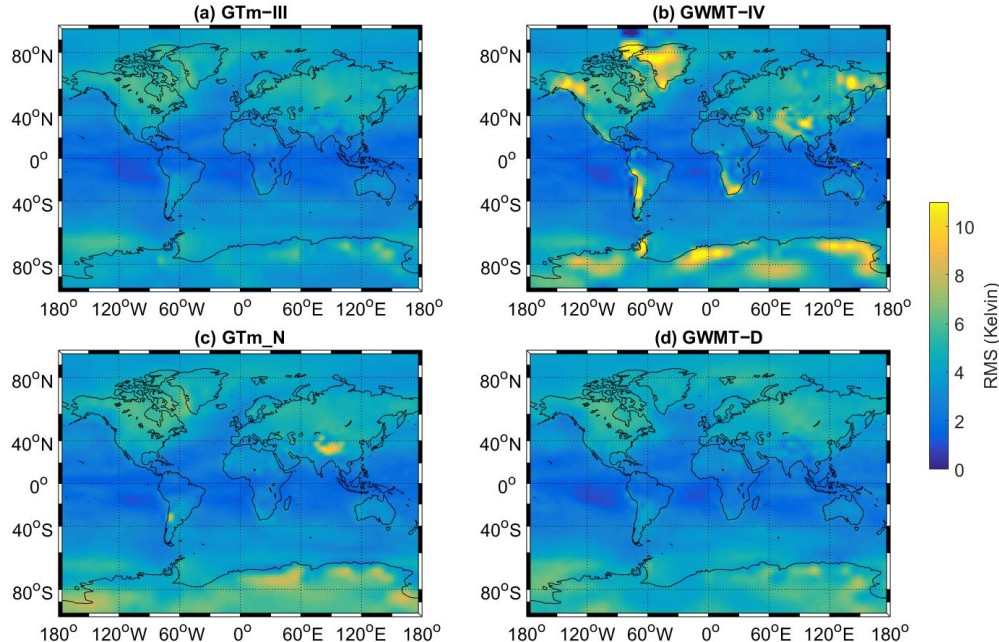

**Figure 7. The global RMS distribution the differences between surface $T_m$ derived from each model and GGOS data in 2014.**

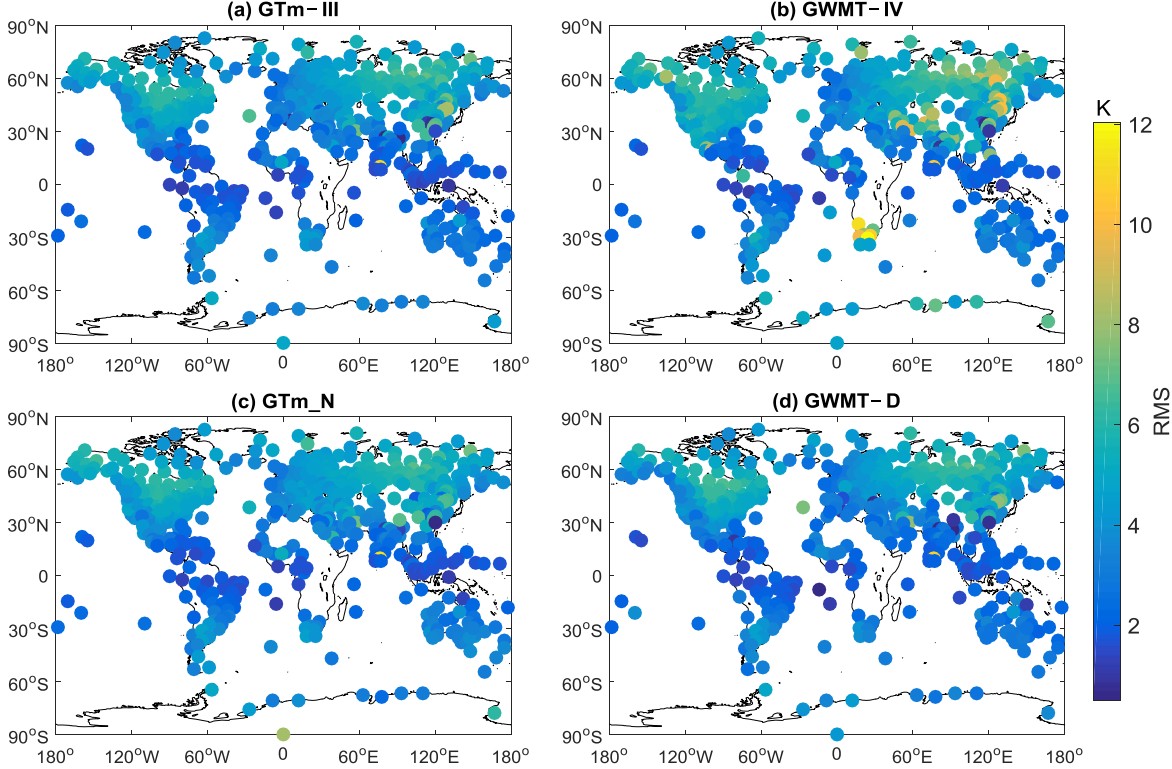

**Figure 8. RMS of model-derived surface $T_m$ in 2014 at 585 global radiosonde sites.**





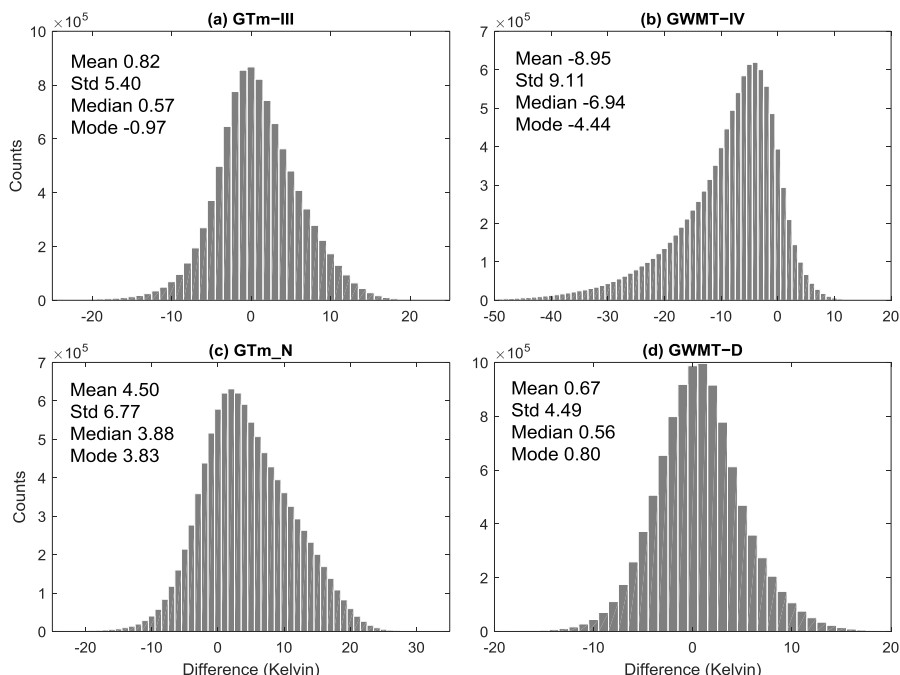

**Figure 9. Histogram of model-derived $T_m$ minus radiosonde-derived $T_m$ in 2014 at different heights.**

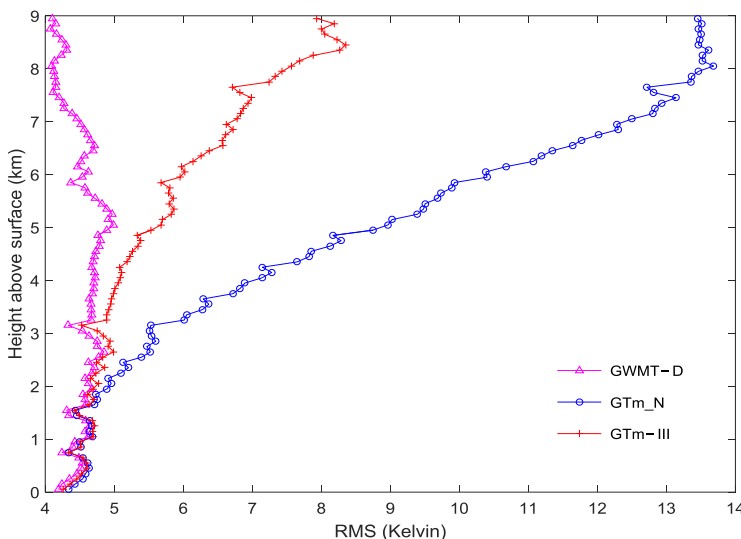

**Figure 10. RMS profile of the $T_m$ from GTm−III, GTm_N, and GWMT−D models, and the observations of 585 radiosonde sites in 2014 are the references.**





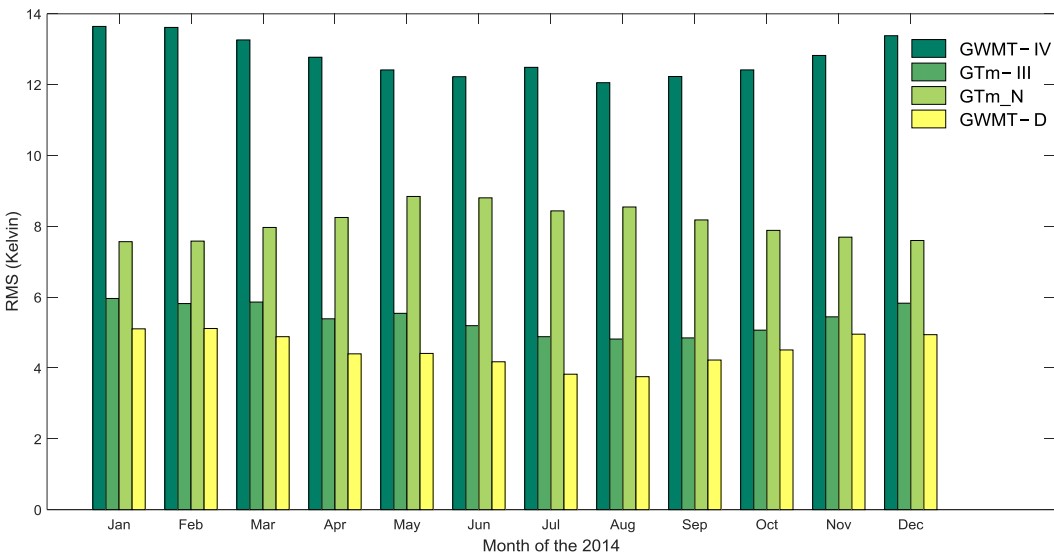

**Figure 11. Monthly-mean RMS of the $T_m$ from the four models and reference values is global radiosonde-derived $T_m$ in 2014.**

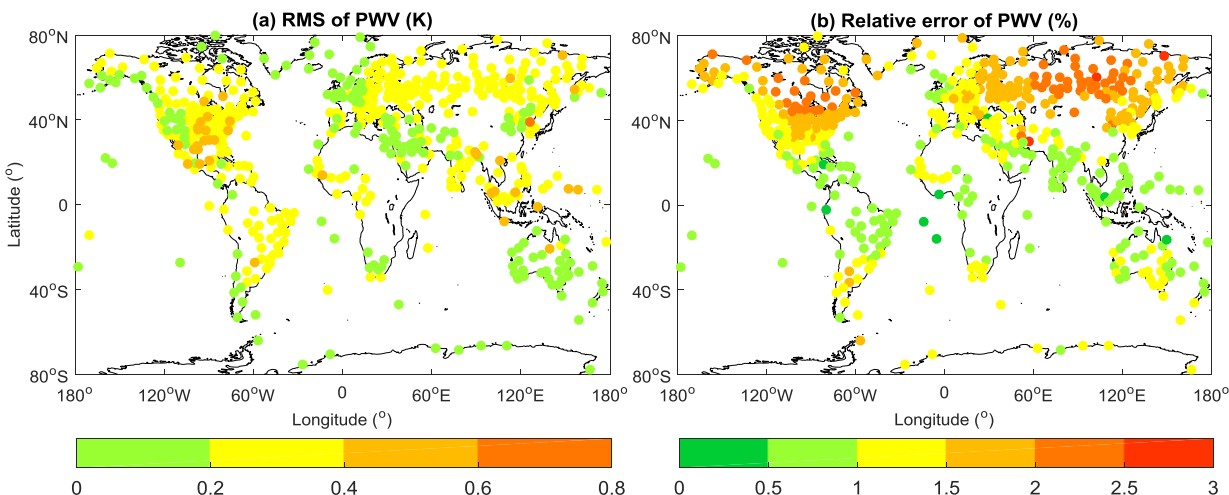

**Figure 12. The theoretical RMS error (a) and relative error (b) of PWV resulting from the GWMT–D model using radiosonde observations in 2014.**

**Table 1.** A list of the latest global empirical $T_m$ models[a].

| Model Name | Feature | Data Source | Input Variable | Surface $T_m$ error (K) | Reference |
|---|---|---|---|---|---|
| UNB3m | Annual | US Standard Atmosphere Supplements | $\varphi$, $\theta$, DOY, $h$ | – | Leandro et al. (2008) |
| GWMT | Spherical Harmonic | Radiosonde (2005−2009) | $\varphi$, DOY, $h$ | 4.6 | Yao et al. (2012) |
| GTm−II | Spherical Harmonic | Radiosonde (2005−2009) | $\varphi$, DOY, $h$ | 4.0 | Yao et al. (2013) |
| GTm−III | Spherical Harmonic | GGOS (2005−2011) | $\theta$, DOY, HOD, $h$ | 4.2 | Yao et al. (2014a) |
| GWMT−IV | Spherical Harmonic | NCEP2(2005−2009) | $\varphi$, DOY, $h$ | ~ 4.1 | He et al. (2013) |
| GTm_N | Spherical Harmonic | NCEP (2006−2012) | $\varphi$, DOY, $h$ | 3.38 | Chen et al. (2014) |
| GTm_X | Grid | ERA-Interim (2007−2010) | $\varphi$, DOY, $h$ | ~ 4.0 | Chen and Yao (2015) |
| GPT2w | Grid | ERA-Interim (2001−2010) | $\varphi$, DOY, $h$ | <4.0[*] | Bohm et al. (2015) |

[a]Their input variables are day of year (*DOY*), hour of day (*HOD*), latitude ($\varphi$), longitude ($\theta$) and surface height ($h$) of a site; the values in the Surface Error column are the RMS of the model on the surface given by the authors, except for the 4.0* of GPT2w, which is a post-fit standard deviation according to the reference).



**Table 2.** The global mean RMS of the differences between the two $T_m$ values derived from GWMT–D built with various lengths of time periods in 2014 NCEP2 at five pressure levels (in K). The values inside square brackets are the minimum and maximum, and the forth row (in bold) are the best fitting results.

| Period length (year) | 1000 hPa | 925 hPa | 850 hPa | 700 hPa | 600 hPa |
|---|---|---|---|---|---|
| 1 | 3.31 [1.16, 12.47] | 3.40 [1.17, 11.66] | 3.50 [1.18, 10.84] | 4.19 [1.16, 9.81] | 4.74 [1.31, 15.38] |
| 2 | 3.24 [1.19, 12.07] | 3.32 [1.17, 11.24] | 3.42 [1.18, 10.40] | 4.13 [1.14, 9.31] | 4.68 [1.30, 14.61] |
| 3 | 3.23 [1.18, 12.33] | 3.32 [1.19, 11.48] | 3.43 [1.21, 10.61] | 4.13 [1.14, 9.47] | 4.67 [1.28, 13.84] |
| **4** | **3.22 [1.15, 11.96]** | **3.32 [1.14, 11.14]** | **3.42 [1.18, 10.34]** | **4.13 [1.14, 9.28]** | **4.67 [1.27, 11.54]** |
| 5 | 3.22 [1.18, 12.13] | 3.31 [1.18, 11.29] | 3.42 [1.18, 10.45] | 4.12 [1.13, 9.37] | 4.66 [2.10, 11.75] |
| 6 | 3.22 [1.17, 12.02] | 3.31 [1.19, 11.20] | 3.42 [1.21, 10.37] | 4.12 [1.14, 9.29] | 4.66 [2.10, 11.75] |
| 7 | 3.22 [1.16, 12.13] | 3.31 [1.19, 11.30] | 3.41 [1.21, 10.47] | 4.11 [1.14, 9.41] | 4.66 [2.10, 13.80] |
| 8 | 3.22 [1.20, 12.16] | 3.31 [1.19, 11.33] | 3.41 [1.21, 10.50] | 4.11 [1.15, 9.43] | 4.66 [1.79, 12.43] |
| 9 | 3.22 [1.20, 12.28] | 3.31 [1.19, 11.42] | 3.41 [1.21, 10.56] | 4.11 [1.15, 9.48] | 4.66 [1.27, 11.55] |





**Table 3.** The globally mean biases and RMSs of the differences between the $T_m$ (in K) derived from four empirical models and 2014 NCEP2 data on pressure levels of 925 hPa and 600 hPa. Values within square brackets are the minimum and maximum, and the % column is the percentage of those global grids with a value $\leq 5$ K.

| Pressure level | Model | Bias | % | RMS | % |
|---|---|---|---|---|---|
| 925 hPa | GTm−III | −1.31 [−5.19, 9.63] | 98.0 | 3.91 [1.26, 15.38] | 77.1 |
| | GWMT−IV | −1.89 [−11.40, 4.77] | 96.2 | 4.36 [1.36, 14.61] | 70.3 |
| | GTm_N | −1.25 [−8.53, 9.18] | 97.1 | 3.84 [1.16, 13.84] | 77.4 |
| | GWMT−D | −0.03 [−2.50, 4.62] | 100 | 3.32 [1.14, 11.14] | 91.1 |
| 600 hPa | GTm−III | −1.25 [−9.30, 4.92] | 89.4 | 5.63 [2.10, 11.75] | 33.2 |
| | GWMT−IV | −5.83 [1.69, 12.35] | 38.4 | 7.28 [2.10, 13.80] | 13.7 |
| | GTm_N | 2.65 [−9.10, 8.81] | 72.9 | 6.38 [1.79, 12.43] | 26.0 |
| | GWMT−D | 0.03 [−2.48, 3.28] | 100 | 4.67 [1.27, 11.54] | 58.3 |


**Table 4.** Global statistics of the differences between the surface $T_m$ derived from four models and GGOS data in 2014 (in K).

| Model | Bias | % | RMS | % |
|---|---|---|---|---|
| GTm−III | −0.02 [−4.44, 4.93] | 100 | 3.29 [0.98, 6.62] | 92.3 |
| GWMT−IV | −0.88 [−20.05, 13.61] | 92.5 | 3.95 [0.91, 20.37] | 76.4 |
| GTm_N | −0.27 [−7.07, 10.09] | 98.2 | 3.70 [1.08, 10.66] | 83.8 |
| GWMT−D | 1.20 [−1.48, 6.23] | 99.5 | 3.54 [0.83, 7.51] | 86.2 |



**Table 5.** Statistics of the differences between model-derived and radiosonde-derived $T_m$ in various height intervals for the year of 2014 at 585 global radiosonde sites (in K).

| Height (km) | % | GWMT−D Bias (RMS) | GTm_N Bias (RMS) | GWMT−IV Bias (RMS) | GTm−III Bias (RMS) |
|---|---|---|---|---|---|
| <2 | 30.1 | 0.52 (4.42) | −0.39 (4.50) | −3.21 (5.20) | −0.73 (4.48) |
| 2~5 | 34.1 | 0.94 (4.67) | 3.23 (6.00) | −8.18 (10.11) | 3.23 (4.82) |
| 5~9 | 35.8 | 0.51 (4.50) | 9.83 (11.55) | − 14.53(18.33) | 9.83 (6.50) |