# Peer review of "A new voxel-based model for the determination of atmospheric-weighted-mean temperature in GPS atmospheric sounding"

_Atmospheric Measurement Techniques, 2016_

## Referee Comment (RC1) · Anonymous Referee #3 · 19 Dec 2016

The submitted manuscript describes the establishment of a voxel-based global weighted mean temperature model, named $GWMT-D$, which is a new version of the GWMT series models. This model obviously improves the modelling performance of weighted mean temperature at higher altitudes compared to the old GWMT models. Such study may be of interested to the community using weighted mean temperature model.

Major comments

01. The primary problem of this manuscript lies in the description and writing. It is found that the methods and explanations in some parts of the manuscript are very difficult to understand. Some descriptions are inaccurate or illogical. The writing of the

manuscript needs great improvement and it is recommended to do this with the help of a professional writer. Some examples are shown below.

Page 5, lines 5-6: "Due to the fact that the GGOS data set has been applied in the development GTm−III, the surface Tm from the GGOS 5 data set is also used in the performance assessment of three selected empirical Tm models." Not logical.

Page 6, lines 8-10: "This paper takes this feature into account and a new modelling procedure is designed to capture the diurnal variation, i.e. Tm values at any other time are obtained by a spline interpolation method." Not clear.

Page 6, lines 26-28: "Long-term Tm time series over the globe can be used for climatological analysis, but its temporal correlation may be too weak to be considered in the Tm modelling process. This suggests that short period of data ma0y lead to an unreliable result." Not logical.

Page 8, lines 18: "Particularly, the constant-value method performs poorly in both temporal and spatial domains."

Page 9, lines 4: "terrain of the Antarctic is generally higher than the pressure level of 1000 hPa"

Page 10, lines 9-11: "Comparing with the GTm_N model, better performance of the GTm−III may result from the discrepancy between GGOS surface Tm data (ECWMF reanalysis data) used by GTm-III and NCEP reanalysis data used by GTm_N." Not understand.

02. In the introduction, the GMWT series models are described in detail, but no introduction about GMWT-IV model is found here. However, in section 4 this model is compared with the new model GMWT-D.

03. Page 2, lines 20-23: Generally, the regression model is also a type of empirical models. Please give a more accurate definition of empirical model.

04. Please explain why the four height levels, 0, 2, 5 and 9 km are chosen in the GWMT-D model.

05. Section 3 is hard to understand. Please rewrite this section and describe the procedure of creating GWMT-D model.

06. The manuscript claims that the study has improved the diurnal variation in the Tm model. However, it is not found any methods on the diurnal variation modelling in the section "3.1.1 Diurnal variation". In addition, the effect of including diurnal variation in the Tm model is also not shown in the manuscript.

07. Page 8, please explain Table 3.

08. Lack of detailed descriptions on the methods used to produce results shown in the Figures 9 and 10 in the manuscript. What is the vertical resolution of Figure 10?

09. Please add the number of radiosonde stations in each height interval to Table 5. However, I think there are no radiosonde stations at the altitudes above 5 km.

Minor comments:

01. Page 3, line 5: "e.g., NCEP-DOE Atmospheric Model Intercomparison 2 (NCEP2) data" ?

02. Page 3, line 27: "2. Data for the determination of Tm" "Data" is enough here.

03. Page 4, line 27: "(5) the highest humidity level is far lower than the height of the top troposphere obtained from an empirical model (200∼350 hPa)" "far lower" is not accurate. Show the accurate height interval.

04. Page 5, line 2: "(i.e. the lower limit of the integral boundary in Eq. (3) is the surface of the site)". Delete it.

05. Page 7, line 23: "four reference times (i.e. from 0, 6, 12, 18, and 24 UTC) of the day" Delete '24'.

06. Page 9, line 15: "modelling method" may be better.

07. Page 13, line 24: "The GTm−II model was identical to GWMT in theory but with different model coefficients" Not clear.

---

## Referee Comment (RC2) · Anonymous Referee #1 · 29 Dec 2016

The manuscript describes a new empirical model (called GWMT-D) for the determination of weighted-mean temperature (Tm). The main focus laid on the modelling of daily variations, the finding of the optimal period for the determination of semi-annual and annual variations from numerical weather model (NWM) data and the reduction of global bias and RMS with respect to radiosonde and NWM data, especially at higher atmospheric levels.

GWMT-D provides mean values, annual, semi-annual and daily variations on a 5° x 5° global grid and four distinct height levels at 0, 2, 5 and 9 km. These parameters were derived from four years of NCEP2 data. Unfortunately it remains unanswered why exactly these height layers were chosen.

Nevertheless, in contrast to other state-of-the-art empirical models based on spherical harmonics (GTm-III, GWMT-IV and GTm_N), the gridded GWMT-D model has a smaller global mean RMS on surface level and on distinct height levels up to 9 km above surface.

From Figure 2 only small Tm variations are visible during daytime. A comparison of daily mean values with daily variations is missing. Thus it is not entirely clear how the modelling of daily variations improves the performance of the model.

In consequence major potential for improvements is seen in the analysis of daily variations and the description of its impact on the model performance. Further the authors should make transparent their decision making process for the selection of the four height levels at 0, 2, 5 and 9 km. In the following I provide some further recommendations and corrections, separately for content and language.

Content:

- Page 2, Line 13 "Rv is the specific gas constant for the air; [. . .]" Rv is the specific gas constant for water vapour

- Page 2, Line 14: "e is the WV pressure (in hPa); [. . .]" Water vapour pressure e does not appear in Eq. 3 but rather density v of liquid water. In order to be consistent with Eq. A1 the ideal gas equation v =e/R*T should be added or at least the relation between e and v should be explained here.

- Page 4, Line 26: Why is the minimum number of valid levels set to 20? The values seems to be too large. In the text above only 17 standard pressure levels are defined.

- Page 5, Line 18: Why did you select exactly these four heights layer for modeling of vertical Tm lapse rate? Please give an inside into the decision making process.

- Page 8, Line 8: "Section 3.1.2 shows that the piecewise linear algorithm [. . .] is better than the direct modeling of Tm [. . .]." In section 3.1.2 the piecewise linear algorithm is mentioned as new model feature. Up to now no results are shown that the piecewise

linear algorithm is better suited than any other approach.

- Page 18, Figure2: The daily variations seem to be rather small, how large is the improvement when 6 hour values are used and interpolated by splines in comparison to daily mean values? Is it worth to add daily variations to the model? Please provide some numbers.

- Page 8, Line 19: Table 2 shows the results only for pressure levels from 1000 hPa to 600 hPa and not below as mentioned in the text.

- Page 12, Eq. A2-A4: Equations not found in given reference Aparicio et al., 2009.

- Page 12, Eq. A3: For the determination of gravity usually a height dependent term like $(1(1+h/RE))^2$ is added. Please explain why this was not used here.

- Page 12 Eq. A5: relative humidity is abbreviated with 'f' but in the text 'RH' is used, please be consistent.

- Page 23, Figure 12: Is the RMS of PWV given in (K) or rather in (mm)?

Understanding and language:

Please review language and writing. A selection of not meaningful or incorrect phrases is given in the following:

- Page 1, Line 8: "One of the most critical variables in PWV remote sensing using GPS technique is the zenith tropospheric delay (ZTD)." Not a good introduction for a paper about mean temperature.

- Page 1, Line 12: "using global reanalysis data from 2010 to 2014 provided by NCEP-DOE Reanalysis 2 data (NCEP2)." Please correct, e.g. in the following way "using global reanalysis data 2 provided by the National Centers for Environmental Prediction (NCEP2)."

- Page 2, Line 1: "using GPS-PWV" can be eliminated

- Page 2, Line 3: “over the site of the station (. . .).” Please clarify

- Page 2, Line 4: “over the region covered by the stations (. . .).” Try to be more precise.

- Page 4, Line 2: The first sentence of Section 2.1 is not meaningful, reanalysis data cannot have a main aim, please correct.

- Page 4, Line 13 “Radiosonde profile data from [. . .] stations over the globe in 2014 (. . .) . . . .”

- Page 5, Line 5: “Due to the fact that . . . the surface Tm from GGOS data is also used . . . .” clarify

- Page 7, Line 4: Assuming Tm at the target location [. . .] is Tm [. . .]” clarify

---

## Author Comment (AC1) · 15 Feb 2017

The submitted manuscript describes the establishment of a voxel-based global weighted mean temperature model, named GWMT−D, which is a new version of the GWMT series models. This model obviously improves the modelling performance of weighted mean temperature at higher altitudes compared to the old GWMT models. Such study may be of interested to the community using weighted mean temperature model.

Major comments

01. The primary problem of this manuscript lies in the description and writing. It is found that the methods and explanations in some parts of the manuscript are very difficult to understand. Some descriptions are inaccurate or illogical. The writing of the manuscript needs great improvement and it is recommended to do this with the help of a professional writer.

*Response: Many thanks. This paper has been revised according to your comments.*

Some examples are shown below.

Page 5, lines 5-6: "Due to the fact that the GGOS data set has been applied in the development GTm−III, the surface Tm from the GGOS data set is also used in the performance assessment of three selected empirical Tm models." Not logical.

*Response: The GTm−III model was developed using GGOS surface Tm. Thus, this model may have a better consistency with GGOS data than the other models which were not based on the GGOS data. This sentence has been rephrased as 'The GGOS data set has been applied in the development of GTm−III and will be also used in the performance assessment of this study' (Page 5 Line 2).*

Page 6, lines 8-10: "This paper takes this feature into account and a new modelling procedure is designed to capture the diurnal variation, i.e. Tm values at any other time are obtained by a spline interpolation method." Not clear.

*Response: The only purpose of this section is to emphasize the significance of adding the diurnal component in Tm modeling. The details of the modeling will be given in Section 3.2. Therefore, this sentence has been rewritten as 'This study takes into account this feature by modelling Tm at each of the four reference times. Whilst, $T_m$ values at any other times can be obtained by the spline interpolation method' (Page 6 Line 5).*

Page 6, lines 26-28: "Long-term Tm time series over the globe can be used for climatological analysis, but its temporal correlation may be too weak to be considered in the Tm modelling process. This suggests that short period of data may lead to an unreliable result." Not logical.

*Response: Agree. These are two parallel viewpoints but lack of linking words. One the one hand, long-term (>10 yrs) Tm time series may be weakly correlated in the time domain (temporal correlation). On the other hand, a short period (< 1 yr) of data may lead to unreliable results. It has been rephrased as 'Long-term $T_m$ data (>10 yrs) over the globe can be used for climatological analysis, but the temporal correlation of $T_m$ time series may be too weak to be considered in $T_m$ modelling. However, a set of short-term $T_m$ data (<1 yr) may be insufficiently for reliable results.' (Page 6 Line 21).*

Page 8, lines 18: "Particularly, the constant-value method performs poorly in both temporal and spatial domains."

*Response: The constant-value method, which was used in other Tm models for modelling Tm lapse rate, is a possible reason for the poor performance of both GTm−III and GTm_N. The sentence has been moved to Conclusion and rephrased as 'The results also confirm that the piecewise linear interpolation of Tm (GWMT−D) is better than the direct modelling of Tm lapse rate (GWMT−IV) and the constant-value method (GTm−III and GTm_N)'(Page 11 Line 7).*

Page 9, lines 4: "terrain of the Antarctic is generally higher than the pressure level of 1000 hPa"

*Response: Rephrased as 'the terrain of Antarctica is generally higher than the height at the 1000 hPa pressure level' (Page 8 Line 23).*

Page 10, lines 9-11: "Comparing with the GTm_N model, better performance of the GTm−III may result from the discrepancy between GGOS surface Tm data (ECWMF reanalysis data) used by GTm-III and NCEP reanalysis data used by GTm_N." Not understand.

*Response: Rephrased as "Compared with the GTm_N model, the better performance of GTm−III may result from the fact that GGOS $T_m$, which was derived from ECMWF reanalysis data, is more consistent with the radiosonde data than the NCEP-derived $T_m$' (Page 9 Line 30).*

02. In the introduction, the GMWT series models are described in details, but no introduction about GMWT-IV model is found here. However, in section 4 this model is compared with the new model GMWT-D.

*Response: Amended by adding an introduction to GWMT-IV on page 3 line 11: 'The GWMT−IV model's $T_m$ lapse rate is a function of geodetic coordinates only'.*

03. Page 2, lines 20-23: Generally, the regression model is also a type of empirical models. Please give a more accurate definition of empirical model.

*Response: The difference between regression and empirical models is not clear-cut. Generally speaking, empirical Tm model in this study is also a type of regression models. To be strict, 'the regression model' in this paper refers to Bevis formula (Tm-Ts relationship). Therefore, 'regression model' used in this paper has been replaced by 'Bevis formula' in case of confusion (Page 2 Line 21).*

04. Please explain why the four height levels, 0, 2, 5 and 9 km are chosen in the GWMT-D model.

*Response: The heights are determined empirically. The reason for choosing the 2 km height is that the atmosphere below 2 km suffers the most from the terrain effect. The height level of 9 km, instead of 10 km, is selected because Tm on the 10 km height may be not a valid number (according to the definition of Tm, zero water vapor pressure above this height leads to 0/0). In addition, we found that the result of the four-layer model is satisfactory, and more complex models do not necessarily significantly improve the modeling results. All of these are the main reasons for the selection of the 0, 2, 5 and 9 km height levels in this study (Page 6 Line 14).*

05. Section 3 is hard to understand. Please rewrite this section and describe the procedure of creating GWMT-D model.

*Response: Amended by polishing. The procedure of developing the new Tm model, which is a multi-dimensional model, is not easy to be well explained. Besides, the separation of relevant figures and tables from the text may also make it difficult to be understood. The general ideas of this section include (1) the types of factors have been considered in GWMT−D and the reasons and (2) the detailed procedure for determining Tm using the new model.*

06. The manuscript claims that the study has improved the diurnal variation in the Tm model. However, it is not found any methods on the diurnal variation modelling in the section "3.1.1 Diurnal variation". In addition, the effect of including diurnal variation in the Tm model is also not shown in the manuscript.

*Response: Amended. Figure 2 has been redrawn only for the first example at latitude 30 °N, longitude 5 °W, height 2 km since the results of the other two examples are similar. Figures 2(a) and 2(b) were kept the same as before. Figure 2(c) shows the range (max – min) of daily $T_m$. Figure 2(d) shows power spectrum density of $T_m$ residuals derived from the GTm–III (using daily-mean method) and GWMT–D (using spline interpolation). Figure 2(d) indicates that GWMT–D effectively captures diurnal variations in $T_m$ but GTm−III does not. Although the post-fitting standard deviations of these two methods are very close (~ 3 K) at all the reference times (0, 6, 12, 18 UTC), the spline interpolation used in GWMT–D can significantly remove the diurnal variation (indirect evidence can be seen from Figure 2(d)). However, this can hardly be validated in this study because all the data used are at the reference times. Other data sets (e.g., COSMIC data) may be used in the future studies (Page 6 Line 1).*

[Figure]

***Figure 2. Statistical results of diurnal $T_m$ (mean ± standard deviation) at 2 km height and four reference times during (a) Dec-Jan-Feb (DJF) and (b) Jun-Jul-Aug (JJA) in 2013, (c) the range (max − min) of daily $T_m$ and (d) power spectrum density (PSD) of $T_m$ residuals.***

07. Page 8, please explain Table 3.

*Response: Table 3 shows the means of the biases and RMSs of all global grid points (with a resolution of ~2 degrees), we can summarize the statistical results on each grid. In order to make it easier to be understood, the sentence on Page 8 Line 9 has been rephrased as 'As a result, the Biases and RMSs of all global grid points on pressure levels of 925 hPa (~0.6 km) and 600 hPa (~5 km) are given in Table 3'.*

08. Lack of detailed descriptions on the methods used to produce results shown in the Figures 9 and 10 in the manuscript. What is the vertical resolution of Figure 10?

*Response: Figure 9 is the histogram of the difference between model-derived Tm and radiosonde-derived Tm at all heights from 0 to 9 km. But this figure cannot reflect the terrain's effect. For example, a radiosonde station on the Tibet plateau has a high altitude but it is very close to the surface. Therefore, Figure 10 is for showing the relationship between the accuracy of Tm models and*

*the height above surface. These two figures together give sufficient information for the procedure. The vertical resolution of Figure 10 is 100 m.*

09. Please add the number of radiosonde stations in each height interval to Table 5. However, I think there are no radiosonde stations at the altitudes above 5 km.

*Response: Amended. Only few stations are located higher than 5 km altitude (e.g. the mountainous area and plateau).* **However, the % column in this table is the percentage of radiosonde records within the height interval in that of all height intervals.** *This description has been added in the title of Table 5 (Page 28).*

Minor comments:

01. Page 3, line 5: "e.g., NCEP-DOE Atmospheric Model Intercomparison 2 (NCEP2) data"?

*Response: This is the full name of the reanalysis project of NCEP. NCEP2 is the official abbreviation and familiar to researchers, thus this abbreviation keep this abbreviation unchanged but move '(NCEP2)' to the end (Page 3 Line 4).*

02. Page 3, line 27: "2. Data for the determination of Tm" "Data" is enough here.

*Response: Amended.*

03. Page 4, line 27: "(5) the highest humidity level is far lower than the height of the top troposphere obtained from an empirical model (200~350 hPa)" "far lower" is not accurate. Show the accurate height interval.

*Response: An empirical model for the height of the tropopause (top layer of the troposphere) used in this study can output the mean height (μ) and standard deviation (σ). If the highest humidity level is less than μ-4σ, this radiosonde data will be removed in the validations. This sentence has been rephrased as ʻthe highest humidity level (200~350 hPa) is less than μ−4σ where μ and σ are the mean height of the top troposphere and its standard deviation obtained from an empirical model' (Page 4 Line 26).*

04. Page 5, line 2: "(i.e. the lower limit of the integral boundary in Eq. (3) is the surface of the site)". Delete it.

*Response: Deleted.*

05. Page 7, line 23: "four reference times (i.e. from 0, 6, 12, 18, and 24 UTC) of the day" Delete '24'.

*Response: Deleted.*

06. Page 9, line 15: "modelling method" may be better.

*Response: Amended.*

07. Page 13, line 24: "The GTm−II model was identical to GWMT in theory but with different model coefficients" Not clear.

*Response: Amended. This sentence has been rephrased as "The GTm−II and GWMT models are developed using the same methodology but with different data".*

---

## Author Comment (AC2) · 16 Feb 2017

The manuscript describes a new empirical model (called GWMT-D) for the determination of weighted-mean temperature (Tm). The main focus laid on the modelling of daily variations, the finding of the optimal period for the determination of semi-annual and annual variations from numerical weather model (NWM) data and the reduction of global bias and RMS with respect to radiosonde and NWM data, especially at higher atmospheric levels.

GWMT-D provides mean values, annual, semi-annual and daily variations on a 5 °×5 ° global grid and four distinct height levels at 0, 2, 5 and 9 km. These parameters were derived from four years of NCEP2 data. Unfortunately it remains unanswered why exactly these height layers were chosen.

Nevertheless, in contrast to other state-of-the-art empirical models based on spherical harmonics (GTm-III, GWMT-IV and GTm_N), the gridded GWMT-D model has a smaller global mean RMS on surface level and on distinct height levels up to 9 km above surface.

From Figure 2 only small Tm variations are visible during daytime. A comparison of daily mean values with daily variations is missing. Thus it is not entirely clear how the modelling of daily variations improves the performance of the model.

In consequence major potential for improvements is seen in the analysis of daily variations and the description of its impact on the model performance. Further the authors should make transparent their decision making process for the selection of the four height levels at 0, 2, 5 and 9 km. In the following I provide some further recommendations and corrections, separately for content and language.

*Response: Many thanks for your comments. The two issues mentioned above will be specifically explained below.*

Content:

- Page 2, Line 13 "Rv is the specific gas constant for the air; [: : :]" Rv is the specific gas constant for water vapour

*Response: Amended.*

- Page 2, Line 14: "e is the WV pressure (in hPa); [: : :]" Water vapour pressure e does not appear in Eq. 3 but rather density v of liquid water. In order to be consistent with Eq. A1 the ideal gas equation v =e/R*T should be added or at least the relation between e and v should be explained here.

*Response: Amended by adding "Using the ideal gas law for the water vapour, $\rho_v$ can be written as $\rho_v = e\,T / R_v$, where e is the WV pressure (in hPa)" (Page 2 Line 15).*

- Page 4, Line 26: Why is the minimum number of valid levels set to 20? The values seem to be too large. In the text above only 17 standard pressure levels are defined.

*Response: These two values refer to two different data or observations. The reanalysis data (NCEP2 in this paper) is provided on the 17 standard pressure levels. The value 20 of valid levels is used in the quality control process of radiosonde observations. In order to remove the effect of the balloon drift and other error sources, the minimum number of valid levels is set to a relatively large value (Page 4 Line 22).*

- Page 5, Line 18: Why did you select exactly these four heights layer for modeling of vertical Tm lapse rate? Please give an inside into the decision making process.

*Response: Generally speaking, the reference heights levels in GWMT−D are determined empirically.*

*The reason for choosing the 2 km height is that the atmosphere below 2 km suffers the most from the terrain effect, and the reason for neglecting 10 km height is the Tm above this level may be not-a-number (zero water vapor above this height lead to 0/0 according to the definition of the Tm, so the Tm cannot be determined for this case). The results show that multi-layer model with more than four layers cannot archieve significantly improvence. Therefore, 0, 2, 5 and 9 km height levels are selected in the GWMT−D model (Page 6 Line 14).*

- Page 8, Line 8: "Section 3.1.2 shows that the piecewise linear algorithm [: : :] is better than the direct modeling of Tm [: : :]." In section 3.1.2 the piecewise linear algorithm is mentioned as new model feature. Up to now no results are shown that the piecewise linear algorithm is better suited than any other approach.

*Response: Amended. I agree. Section 3.1.2 only shows the horizontal variation of Tm lapse rate. Nevertheless, the results in Section 4 Validation of Tm models confirm that "The results also show that the piecewise linear interpolation of $T_m$ used in GWMT−D is better than the direct modelling of $T_m$ lapse rate in GWMT−IV or the constant-value method used in both GTm−III and GTm_N.". Therefore, this sentence has been moved to the Conclusion section (Page 11 Line 7).*

- Page 18, Figure2: The daily variations seem to be rather small, how large is the improvement when 6 hour values are used and interpolated by splines in comparison to daily mean values? Is it worth to add daily variations to the model? Please provide some numbers.

*Response: Amended. Figure 2 has been redrawn only for the first example at latitude 30 °N, longitude 5 °W, height 2 km since the results of the other two examples are similar. Figures 2(a) and 2(b) were kept the same as before. Figure 2(c) shows the range (max − min) of daily $T_m$. Figure 2(d) shows power spectrum density of $T_m$ residuals derived from the GTm–III (using daily-mean method) and GWMT–D (using spline interpolation). Figure 2(d) indicates that GWMT–D effectively captures diurnal variations in $T_m$ but GTm−III does not. Although the post-fitting standard deviations of these two methods are very close (~ 3 K) at all the reference times (0, 6, 12, 18 UTC), the spline interpolation used in GWMT–D can significantly remove the diurnal variation (indirect evidence can be seen from Figure 2(d)). However, this can hardly be validated in this study because all the data used are at the reference times. Other data sets (e.g., COSMIC data) may be used for the future work (Page 6 Line 1).*

[Figure]

***Figure 2. Statistical results of diurnal $T_m$ (mean ± standard deviation) at 2 km height and four***

*reference times during (a) Dec-Jan-Feb (DJF) and (b) Jun-Jul-Aug (JJA) in 2013, (c) the range (max − min) of daily $T_m$ and (d) power spectrum density (PSD) of $T_m$ residuals.*

- Page 8, Line 19: Table 2 shows the results only for pressure levels from 1000 hPa to 600 hPa and not below as mentioned in the text.

*Response: Amended. In fact, the reasons were explained Section 4.1. In order to make it consistent, this part in Section 4.1 has been moved from Section 4.1 to Section 3.1.3 and rephrased as 'As a result, the Bias and RMS of all global grid points are given in Table 3 on the pressure levels of 925 hPa (~0.6 km) and 600 hPa (~5 km).' (Page 6, Line 29).*

- Page 12, Eq. A2-A4: Equations not found in given reference Aparicio et al., 2009.

*Response: Amended by replacing with "(Ge, 2006)".*

- Page 12, Eq. A3: For the determination of gravity usually a height dependent term like (1(1+h/RE))^2 is added. Please explain why this was not used here.

*Response: Amended. $g(\varphi)$ is in fact the gravity on the geoid which has been added in the descriptions of Equation (A3). Traditionally, the geopotential height is defined based on a reference height which is usually the The dependence of height in gravity has been considered in the derivation of Equation (A2). More details can refer to Ge (2006).*

- Page 12 Eq. A5: relative humidity is abbreviated with 'f' but in the text 'RH' is used, please be consistent.

*Response: Amended. 'f' in the Eq. A5 is an enhancement factor defined as the ratio of the saturation vapour pressure of moist air to that of pure water vapour (WMO, 2000), so f is a coefficient calibrating the results for ideal water vapour to moist air. The 'f' in Equation (A5)−(A8) have been replaced by 'f(P)' for clarification, e.g., $e = f(P)exp\left(\frac{17.62\ t}{243.12+t}\right)$.*

- Page 23, Figure 12: Is the RMS of PWV given in (K) or rather in (mm)?

*Response: Amended. The unit in Figure 12(a) is mm.*

Understanding and language:

Please review language and writing. A selection of not meaningful or incorrect phrases is given in the following:

- Page 1, Line 8: "One of the most critical variables in PWV remote sensing using GPS technique is the zenith tropospheric delay (ZTD)." Not a good introduction for a paper about mean temperature.

*Response: Amended as 'In the GPS-based PWV remote sensing, the atmospheric-weighted-mean temperature (Tm) is a crucial parameter for the conversion from zenith tropospheric delay (ZTD) to PWV over the GPS station' (Page 1 Line 8).*

- Page 1, Line 12: "using global reanalysis data from 2010 to 2014 provided by NCEP-DOE Reanalysis 2 data (NCEP2)." Please correct, e.g. in the following way "using global reanalysis data 2 provided by the National Centers for Environmental Prediction (NCEP2)."

*Response: Amended.*

- Page 2, Line 1: "using GPS-PWV" can be eliminated

*Response: Amended.*

- Page 2, Line 3: "over the site of the station (: : :)." Please clarify

*Response: Amended as 'The GPS-PWV have been used to study the temporal variation of PWV, such as seasonal and diurnal variation patterns'.*

- Page 2, Line 4: "over the region covered by the stations (: : :)." Try to be more precise.

*Response: Amended as 'It also has been used to investigate the spatial variation in PWV over the GPS network'.*

- Page 4, Line 2: The first sentence of Section 2.1 is not meaningful, reanalysis data cannot have a main aim, please correct.

*Response: The numerical weather prediction/analysis (NWP) system project usually aims at improving NWP models, or providing atmospheric dataset for studies including, e.g., climate change and monitoring and numerical seasonal prediction. There are some papers which introduce the motivation of reanalysis data (e.g., NCEP2, ECMWF ERA-Interim, and JRA-55).*

*(1) Kanamitsu, M., Ebisuzaki, W., Woollen, J., Yang, S. K., Hnilo, J. J., Fiorino, M., and Potter, G. L.: NCEP-DOE AMIP-II reanalysis (R-2), Bulletin of the American Meteorological Society, 83, 1631-1643, 10.1175/Bams-83-11-1631, 2002.*

*(2) Ebita, A., Kobayashi, S., Ota, Y., Moriya, M., Kumabe, R., Onogi, K., & Kamahori, H.: The Japanese 55-year Reanalysis" JRA-55": an interim report.Sola,7, 149-152, 2011.*

*In order to avoid further misunderstanding here, this sentence has been rephrased as 'The studies of climate change and climate monitoring benefit from the National Centers for Environmental Prediction/National Center for Atmospheric Research (NCEP/NCAR) reanalysis data' (Page 4 Line 2).*

- Page 4, Line 13 "Radiosonde profile data from [: : :] stations over the globe in 2014 (: : :) : : : ."

*Response: Amended as 'Radiosonde profiles from 585 global Integrated Global Radiosonde Archive (IGRA) stations (Figure 1) are selected to validate the new GWMT−D model'.*

- Page 5, Line 5: "Due to the fact that : : : the surface Tm from GGOS data is also used : : : ." clarify

*Response: Amended as 'The GGOS data set has been applied in the development of GTm−III and will be used in the performance assessment of this paper'.*

- Page 7, Line 4: Assuming Tm at the target location [: : :] is Tm [: : :]" clarify

*Response: Amended as 'Assuming $T_m$ (φ, λ, h, DOY, HOD) is a function of target location (φ, λ, h), day of year (DOY) and UTC hour (HOD),…'.*